# Assessment of Adult Patients with Long COVID Manifestations Suspected as Cardiovascular: A Single-Center Experience

**DOI:** 10.3390/jcm11206123

**Published:** 2022-10-18

**Authors:** Alon Shechter, Dana Yelin, Ili Margalit, Merry Abitbol, Olga Morelli, Ashraf Hamdan, Mordehay Vaturi, Alon Eisen, Alex Sagie, Ran Kornowski, Yaron Shapira

**Affiliations:** 1Department of Cardiology, Rabin Medical Center, Petach Tikva 4941492, Israel; 2Faculty of Medicine, Tel Aviv University, Tel Aviv 6910203, Israel; 3Long-COVID Clinic, Rabin Medical Center, Petach Tikva 4941492, Israel

**Keywords:** long COVID, cardiovascular complications, prognosis

## Abstract

*Background*: Persistent symptoms affect a subset of coronavirus disease 2019 (COVID-19) survivors. Some of these may be cardiovascular (CV)-related. *Objective*: To assess the burden of objective CV morbidity among, and to explore the short-term course experienced by, COVID-19 patients with post-infectious symptomatology suspected as CV. *Methods*: This was a single-center, retrospective analysis of consecutive adult patients with new-onset symptoms believed to be CV following recovery from COVID-19, who had been assessed at a dedicated ‘Cardio’-COVID clinic between June 2020 and June 2021. All participants were followed for 1 year for symptomatic course and the occurrence of new CV diagnoses and major adverse cardiovascular events (MACE). *Results*: A total of 96 patients (median age 54 (IQR, 44–64) years, 52 (54%) females) were included in the final analysis. Initial visits occurred within a median of 142 days after the diagnosis of acute COVID. Nearly all (99%) patients experienced a symptomatic acute illness, which was graded as severe in 26 (27%) cases according to the National Institutes of Health (NIH) criteria. Long-COVID symptoms included mainly dyspnea and fatigue. While the initial work-up was mostly normal, 45% of the 11 cardiac magnetic resonance studies performed revealed pathologies. New CV diagnoses were made in nine (9%) patients and mainly included myocarditis that later resolved. An abnormal spirometry was the only variable associated with these. No MACE were recorded. Fifty-two (54%) participants felt that their symptoms improved. No association was found between CV morbidity and symptomatic course. *Conclusions*: In our experience, long-COVID symptoms of presumed CV origin signified actual CV disease in a minority of patients who, irrespective of the final diagnosis, faced a fair 1-year prognosis.

## 1. Introduction

Shortly after its worldwide outbreak in early 2020, numerous reports have been documented that link coronavirus disease 2019 (COVID-19) with various cardiovascular (CV) manifestations including myocarditis [1,2,3], thromboembolic events [4], arrhythmias [5,6,7], and takotsubo cardiomyopathy [8,9]. Endothelial dysfunction [10,11], cytokine storm [12,13], hypoxia-induced injury [14], and direct myocardial infiltration by the virus [15,16] are few of the mechanisms offered as possible explanations for the association. Moving forward, and upon the mass conclusion of the disease’s acute phase, a shift to a so-called ‘long’ (formerly ‘post’)-COVID has been observed, which, depending on its exact definition and population sampled, may affect a non-negligible portion of survivors [17,18]. As suggested by previous works, symptoms of presumed CV origin are not uncommon in this chronic condition [19,20,21,22]. Huang et al., for instance, in their pioneering large-scale report, observed dyspnea on exertion, palpitations, and chest pain in 24%, 10%, and 5%, respectively, of 1,733 patients 6 months after discharge from a Wuhan hospital [23]. Similarly, a Spanish registry of 139 health care workers diagnosed with COVID found a 42% prevalence of CV symptoms some 6–10 weeks post diagnosis [24]. The question remains, however, as to the extent of CV morbidity and course in long-COVID patients with presumed CV symptomatology. To address the matter, a dedicated ‘Cardio’-COVID clinic was initiated at Rabin Medical Center (RMC), Beilinson Hospital (Petach Tikva, Israel), by early June 2020 that accepted adult (i.e., 18 years of age or older) outpatients from across the country who manifested ongoing signs and/or symptoms suspected by their treating physicians to represent CV sequelae while recovering from COVID-19. Herein, we report on the experience of this clinic.

## 2. Methods

### 2.1. Study Design and Participants

This is a retrospective study based on the RMC Cardio-COVID Clinic population described above. Participating individuals were required to: (a) receive a polymerase chain reaction (PCR)-confirmed diagnosis of the disease; (b) be declared as recovered by means of a negative repeat PCR; (c) exhibit symptoms not known to affect them before COVID that lasted at least 60 days after formal recovery; and (d) possess a written referral from their treating physicians stating the exact manifestation(s) thought to imply CV disease. All primary care physicians were certified by Israel’s Ministry of Health in the field(s) of internal medicine and/or family medicine. Aiming to report a full year of follow-up, and in light of the gradual reduction in the clinic’s activity, this report relates to patients seen between June 2020 and June 2021. A schematic presentation of the study construct is provided in Figure 1.

The study was conducted in accordance with the Declaration of Helsinki and received an Institutional Review Board (IRB) approval (numbered 0656-21-RMC).

### 2.2. Assessment and Follow-Up

A uniform baseline evaluation was performed on all participants and consisted of history taking and physical examination by a single cardiologist, electrocardiogram (ECG), and transthoracic echocardiogram (TTE). Functional status was assessed using the New York Heart Association (NYHA) classification. Additional, more advanced exams were utilized as deemed appropriate by the cardiologist and included any of the following: cardiac provocation test, Holter ECG, cardiopulmonary stress test (CPET), cardiac computed tomography (CCT), cardiac magnetic resonance (CMR), pulmonary function test (PFT), and high-resolution CT (HRCT) of the chest. While echocardiography-based studies were read by the clinic’s cardiologist, other studies were interpreted by experts outside of the clinic; in either case, accepted guidelines were followed such as those published by the American Society of Echocardiography [25].

Initial and follow-up data regarding the patients’ medical status including test results and any adverse events were collected either in-person or remotely using Ofek Software (dbMotion, Pittsburg, PA, USA), which is a web-based medical chart program shared by most of Israel’s public medical providers. Details about acute COVID, and particularly hospitalization course, were electronically retrieved in a similar fashion. Acute COVID severity was retrospectively determined using the National Institutes of Health (NIH) criteria [26]. Accordingly, a moderate illness involved any clinical or radiological sign of lower respiratory tract disease without significant desaturation. Conversely, patients with severe COVID had an oxygen saturation of <94% on room air, a ratio of arterial partial pressure of oxygen to fraction of inspired oxygen of <300 mmHg, a respiratory rate of >30 breaths/min, or lung infiltrates >50%. Any new CV diagnosis made during the study period and not clearly explained by another disease state after a comprehensive chart review was regarded as potentially COVID-related.

### 2.3. Outcomes and Exposures

Outcomes of interest included the occurrence of new, potentially COVID-related CV diagnoses made during 1 year of follow-up as well as of major adverse cardiovascular events (MACE), defined as acute coronary syndrome, acute stroke, and CV death. Qualitative symptomatic status, as dictated during history taking by primary care givers or the clinic’s cardiologist, was also recorded. Exposures included all baseline medical conditions as well as objective findings revealed during work-up.

### 2.4. Statistical Analysis

Data are presented as mean ± standard deviation, median (interquartile range), or number (percentage), where appropriate. Categorical variables were compared using the Chi-square or Fisher’s exact tests; continuous variables were compared using the Student’s t test or Mann–Whitney U test for normally or non-normally distributed parameters, respectively.

Based on the presence of new CV diagnoses believed to represent a consequence of COVID as well as the persistence of long-COVID symptoms, the cohort was split into two groups for the purpose of further analyses. A univariate binary logistic regression analysis was utilized to identify possible predictors for new, potentially COVID-related CV diagnoses as well as for symptom non-improvement. Later, parameters demonstrating a *p*-value of <0.1 were integrated into a multivariate model. Considering the sample size and number of events, this model was regarded as exploratory only.

A 2-sided *p*-value of <0.05 was considered for statistical significance. All analyses were performed using Statistical Analysis Software (SAS), Version 9.4 (SAS Institute, Cary, NC, USA), and SPSS Statistics for Windows software, version 28 (IBM Corporation, Armonk, NY, USA).

## 3. Results

A total of 96 patients were included in the final analysis, representing 100% of the clinic’s original cohort. Eighty (83%) participants were referred by RMC Long-COVID Clinic, followed by family physicians (*n* = 12, 13%) and emergency medicine doctors (*n* = 4, 4%). Initial visits occurred within a median of 142 (IQR, 111–197) days after COVID diagnosis. In all, 135 encounters were performed. Seventy (73%) patients were seen in-person once.

### 3.1. Baseline Characteristics

Baseline characteristics of the study population are presented in Table 1. A little more than half of the participants (*n* = 52, 54%) were female, and the median age was 54 (IQR, 44–64) years. Fifteen (16%) patients had prior CV conditions and 69 (72%) had at least one atherosclerotic CV disease risk factor, mostly dyslipidemia (58%) and pre-diabetes/diabetes (35%).

Regarding acute COVID, all participants but one (*n* = 95, 99%) experienced a symptomatic illness. Based on the NIH criteria [26], acute COVID was classified as mild, moderate, or severe in 42%, 31%, and 27% of cases, respectively. Twenty-eight (29%) patients were hospitalized due their disease, for a median duration of 8 (IQR, 5–18) days. Of these, one patient was invasively ventilated. During their hospital stay, 10 out of 22 (45%) patients had ECG aberrations (almost entirely non-specific ST-T changes), two in eight (25%) patients showed TTE findings (mainly LV systolic dysfunction), and three of 23 (13%) patients exhibited elevated levels of serum biomarkers (i.e., high-sensitivity cardiac troponin ± N-terminal-pro-brain natriuretic peptide). No data were available regarding the exact value of cardiac troponin levels that were considered within normal limits by the measuring laboratory.

### 3.2. Long-COVID Manifestations and Findings

Late symptoms not known to affect patients prior to the infectious disease mainly included dyspnea (56%), fatigue (50%), chest pain (42%), and palpitations (38%) (Table 2). Fatigue was universally accompanied by other symptoms, mostly dyspnea (68%) and palpitations (46%). NYHA functional classes I–II and III were observed in 84 (88%) and 12 (12%) patients, respectively. No cases of re-infection were reported.

A schematic presentation of objective findings revealed during patient work-up is provided in Figure 2 and Figure 3. Of note, initial physical examination was normal in all but three (3%) participants who exhibited midsystolic murmurs confined to the upper right sternal border. ECG aberrations were found in 32 (34%) patients—mostly non-specific ST-T changes (21%) and conduction anomalies (13%)—and TTE findings were discovered in 31 (32%) patients—mostly minimal (*n* = 12, 13%) and minimal to mild (*n* = 6, 6%) pericardial effusion; visually-estimated right ventricular function was reduced in one patient (1%). As for advanced tests, the highest rates of abnormal findings were reported for HRCTs (52%), CMRs (45%), and CPETs (42%). A third of PFTs and almost a quarter (23%) of CCTs also produced abnormal results. Out of the 39 patients that were tested for age-adjusted CV fitness, 15 (38%) exhibited good performance, 14 (36%) had an average one, and 10 (26%) showed lower-than-expected fitness. Notably, three (60%) and two (40%) of the five pathologic CMRs were preceded by normal ECGs and TTEs, respectively. Additionally, only one case of CV restraint per CPET translated to provocation test anomalies, and one to the CMR findings.

### 3.3. Cardiovascular Morbidity

Overall, 15 overt CV diagnoses were made in 14 (15%) patients. These included: myocarditis (3), myopericarditis (2), chronotropic incompetence (3), inappropriate sinus tachycardia (1), atrial fibrillation (AF) (1), atrial tachycardia (AT) (1), non-sustained ventricular tachycardia (NSVT) (1), hypertrophic obstructive cardiomyopathy (HOCM) (1), non-obstructive coronary artery disease (NOCAD) (2), and an aberrant coronary artery (ACA) (1). Excluding the last three diagnoses (HOCM, NOCAD, and ACA), which represent pre-infectious states and are thus unrelated to COVID, as well as the single AF case, which was found to long predate the acute infection, and the one NSVT case, which occurred in the setting of a pre-existing ischemic heart disease, nine (9%) patients received never-before known, potentially COVID-related, CV diagnoses. No MACE were documented. All myocarditis and chronotropic incompetence findings disappeared on appropriate subsequent exams (i.e., CMR and provocation test, respectively). Repeat Holter ECG, performed within a median of 3 months of the index study, revealed complete resolution in the single atrial tachycardia and the single inappropriate sinus tachycardia cases. AF, NSVT, and couplet VPBs remained unchanged.

### 3.4. Symptomatic Course

As shown in Figure 4, by the end of the 1-year follow-up period, 52 (54%) patients reported symptomatic improvement, either to their treating physicians or when re-visiting the Cardio-COVID clinic. Two patients felt their condition to deteriorate. Interestingly, dizziness/vertigo was the only symptom that did not improve at all. These trends were shared by those who received new CV diagnoses and those who did not (*p* = 0.816).

### 3.5. Predictors of Presumably COVID-Related New CV Diagnoses and Symptom Non-Improvement

Most (60%) participants with a myocardial injury-related diagnosis suffered a severe acute COVID, and all those with chronotropic incompetence recovered from a moderate one. Apart from LA dilation, which was more common (>4 times more likely to occur) among patients with new CV diagnoses (2/9 vs. 4/87, *p* = 0.017), and abnormal PFT, which was almost so as well (6/9 vs. 21/72, *p* = 0.054), no major differences were noted between the ‘new CV diagnosis’ group (*n* = 9) and the ‘no new CV diagnosis’ group (*n* = 87) (Table 3 and Table 4). As per an exploratory multivariate analysis, shown in detail in Table 5 and Table 6, abnormal PFT was the only parameter to demonstrate a significant independent association with new CV diagnoses (OR 5.16, 95% CI 1.12–23.68, *p* = 0.035). This finding faded upon inspection of specific spirometrical aberrations (obstruction—OR 13.19, 95% CI 0.88–196.96, *p* = 0.061; restriction—OR 0.74, 95% CI 0.05–10.39, *p* = 0.823; reduced diffusion capacity—OR 1.12, 95% CI 0.11–11.85, *p* = 0.927). No similar associations were found for baseline patient characteristics, acute COVID data parameters, vaccination status, long-COVID symptoms or symptomatic course, or other objective findings revealed by the work-up.

Regarding symptomatic status, NYHA functional class proved more advanced in patients whose symptoms did not improve (*p* = 0.008). This was paralleled by a significantly higher proportion of symptomatic non-improvement among patients in NYHA class III compared to patients in NYHA class I–II (9/12;75% vs. 35/84;42%, *p* = 0.030). Additionally, pericardial effusion tended to present more frequently among these participants (12/44 vs. 6/52, *p* = 0.060). All other variables explored, including the very presence (or absence) of a new CV diagnosis, were comparable in those with and those without symptomatic non-improvement (Table 7 and Table 8). Here, a binary logistic regression analysis failed to reveal any statistically significant associations between the symptomatic trend and baseline characteristics (including NYHA class), COVID parameters, assessment results, or overt CV diagnoses (Table 5 and Table 6).

## 4. Discussion

To the best of our knowledge, this is the first study to report on the experience of a dedicated, CV-oriented clinic that specifically cares for COVID survivors with ‘CV’ symptoms. Its main findings can be summarized as follows: (1) Persistent symptoms of possible CV nature after formal recovery from COVID were mostly less-specific (such as dyspnea and fatigue); (2) during one year of follow-up, actual new-onset, presumably COVID-related CV conditions were revealed in a fraction of suspected cases and mainly included self-limiting myocardial disorders and benign supraventricular arrhythmias; (3) irrespective of the presence of a new CV diagnosis, CV prognosis by the end of the 1-year follow-up was fairly benign, with most symptoms improving and all cases of myocarditis and chronotropic incompetence resolving, and without any occurrence of MACE; (4) post-COVID abnormal spirometry was associated with overt CV disease and advanced NYHA class occurred more frequently among patients whose symptoms did not improve, however, no obvious predictors were identified for symptomatic non-improvement; and (5) carefully chosen, CMR and CPET had the highest diagnostic yield.

Our study highlights a discrepancy between subjective CV symptomology, at the very least as perceived by primary care physicians, and significant objective CV morbidity among long-COVID patients. In addition to the low incidence (<10%) of ‘formal’ CV diagnoses made during follow-up, there appeared to be only minor aberrations on the initial work-up tests (including ECG and TTE). Furthermore, no clear association was demonstrated between the presence of such diagnoses and findings and the patients’ symptomatic course. Therefore, while some CV conditions might have been missed, due to diagnostic tests under-/non-utilization or malperformance or disease self-resolution, it is our notion that most patients experienced ‘extra’-cardiac (arguably pulmonary) phenomena in the first place, and/or as yet a poorly defined deconditioning process. Consistent with this assumption is the observed high percentage of abnormal HRCTs (52%, exceeding all CV tests) and PFTs (33%). Although no data were available regarding pre-COVID pulmonary status that could serve as a reference, this figure was roughly similar among smokers and non-smokers, thus pointing to a possible link to the infectious process. The above hypothesis corresponds to a published review of long-COVID that suggested a predominance of chronic residual respiratory aberrations [27]. Pulmonary work-up, and appropriate treatment where necessary, is therefore warranted in COVID survivors with suspected CV symptoms, alongside any cardiology consultation.

In accordance with previous reports [28,29] and an extensive meta-analysis [30] concerning the long-COVID population in general, our study revealed a rather high burden of acute illness among those patients suspected to sustain late CV sequelae. As outlined, acute COVID in our cohort was almost universally symptomatic, led to hospitalization in over a quarter of cases, and was accompanied by abnormal objective findings in nearly half. Moreover, a great majority of patients with myocarditis recovered from a severe acute COVID. Hence, the initial infection may play a crucial role in determining the chronic, post-infectious phase, be it cardiac or non-cardiac in nature, and should probably dictate the extent of the work-up. Equally important is the post recovery functional status, which in our study proved lower among patients whose symptoms did not improve, thus potentially forecasting the duration of late symptoms and consequently needed follow-up.

In regard to the potentially COVID-related, overt CV conditions that were diagnosed in our study, it is somewhat reassuring that COVID may follow the path of other well-known respiratory viral illnesses such as influenza, in its predilection to rather benign, self-limiting myocardial and arrhythmic phenomena [31]. Notably, none of the myocarditis cases involved extensive late gadolinium enhancement on CMR, and all resolved on subsequent scans, much in the same manner others have demonstrated most CMR findings to regress over time in subjects with acute (viral) myocarditis [32]. Similarly, all patients with newly diagnosed chronotropic incompetence experienced full remission, and most of them reported symptomatic improvement. The exact timeframe by which relief could be expected to occur was not explored in the study.

As mentioned, abnormal spirometry was shown by our exploratory analysis to be associated the emergence of new, overt CV diagnoses among long-COVID patients with suspected CV symptoms. As this observation may represent the effect of mere chance, considering the sample size, it could also express a genuine interaction between pulmonary function and CV state. For this matter, an ARIC sub-study of a cohort harboring a similar baseline profile to that of our clinic’s population reported increased rates of incident CV disease among participants with deteriorated lung function [33]. A heightened inflammatory response was cited as a possible mechanistic connection, as suggested by the higher levels of serum high-sensitivity C-reactive protein (CRP) observed in those with impaired spirometry. Although plausible in theory in our study as well, we cannot ascertain a cause-and-effect relationship as most patients did not have any records of pre-COVID pulmonary examinations and because no routine testing of inflammatory markers was performed. This, in addition to the exploratory nature of the current analysis, makes our finding a hypothesis-generating one, that is, until further knowledge becomes available.

Our inability to firmly assign any predictors for long-COVID course as well as the somewhat contradicting results of diagnostic tests performed in search of CV disease, may all stem from the study’s low power, or could alternatively highlight the incomplete understanding of the disease at present. In this respect, while current knowledge assigns a lower likelihood for viral myocarditis in the case of normal baseline ECG and TTE, our findings – including the relative low sensitivity of these tests (60% and 40%, respectively) – suggest symptomatic long-COVID-related myocarditis to be rather clinically subtle and/or subacute (sometimes even chronic) in nature, so as to manifest merely symptomatically and by virtue of CMR. Accordingly, more emphasis probably should be put on clinical judgement. As the assessment of the clinic’s patients was indeed clinically-driven, with no formal guidance to rely upon, some patients were referred to CMR in an early stage of the work-up, keeping in mind that negative results on basic tests may not rule-out the possibility of myocardial injury. It is our hope that future research will aid in defining clinical pathways in the long-COVID arena.

### Limitations

In addition to the implications of the study’s single-center design, and of its limited size and follow-up period, there were several additional important limitations. First, as our analysis focused on long-COVID patients believed to suffer CV sequelae, the study’s results cannot be generalized to the entire long-COVID population nor to unselected COVID survivors, making it impossible to estimate the burden, course, and predictors of CV morbidity and symptomatology in these larger populations. Moreover, our real-time working diagnosis of long-COVID that was used to triage referrals is one of several and should be taken into consideration when interpreting the data. Second, selection and/or survivor biases may have taken place in shaping the study’s cohort, as might be suspected considering the rather young age and lower-than-expected baseline CV morbidity observed, especially among participants with newly-diagnosed CV conditions. It could be that older, sicker long-COVID patients were referred to, or chose to attend, more ‘traditional,’ non-COVID centered, health care establishments. Third, much of the clinic’s routine was based on qualitative measures, starting with the much subjective inclusion criteria (largely left to the discretion of the referring physicians), and continuing with the patients’ assessment, which relied on clinical judgement and that did not systematically incorporate validated questionnaires or scales (other than the NYHA classification of functional status) or serum biomarkers. Furthermore, advanced tests were not universally applied to all patients, but rather tailored to their symptoms. A quantitative, more evidence-based working scheme, as well as routine referral to all available studies regardless of clinical presentation, might have produced different results, and specifically led to a higher number of CV diagnoses. However, as mentioned, our pioneering clinic operated in the midst of the pandemic with no peer-reviewed evidence to serve as guidance, forcing subjective assessment to dictate downstream evaluation. Additionally, aiming to make our results applicable to a real-world, funds-limited setting, we assessed patients in the way we would approach any patient with a suspected CV condition, that is, by exercising clinical judgement. For this reason, we also allowed for broad inclusion criteria, namely the primary care physicians’ perception of possible CV disease, as would be expected to take place in ordinary cardiology consultation services. Importantly, the decision to include patients according to symptoms was taken in the hope of promoting a high level of suspicion on the part of primary care physicians, knowing that CV disease, COVID-related included, may present with non-specific manifestations (some of which reflect low-flow states) such as fatigue. Finally, again reflecting a real-world practice, the patient evaluation was performed by non-blinded staff members; however, other than the clinic’s cardiologist, none of them were exposed to the entire medical charts nor to the test results outside of their specific expertise.

## 5. Conclusions

In this single-center observational study, presumed CV symptoms of long-COVID were accompanied by actual CV disease in a minority of patients. Irrespective of the presence or absence of overt CV conditions, 1-year symptomatic course was fair, with more than half of patients reporting improvement, and no MACE occurred. These data may assist clinicians facing long-COVID patients with ‘CV’ symptomatology when communicating the disease and when deciding upon its assessment. Further research is needed to validate the study’s findings, to identify predictors for long-COVID CV outcome, particularly in patients with suspected late CV sequelae, and to define appropriate clinical pathways.

## Figures and Tables

**Figure 1 jcm-11-06123-f001:**
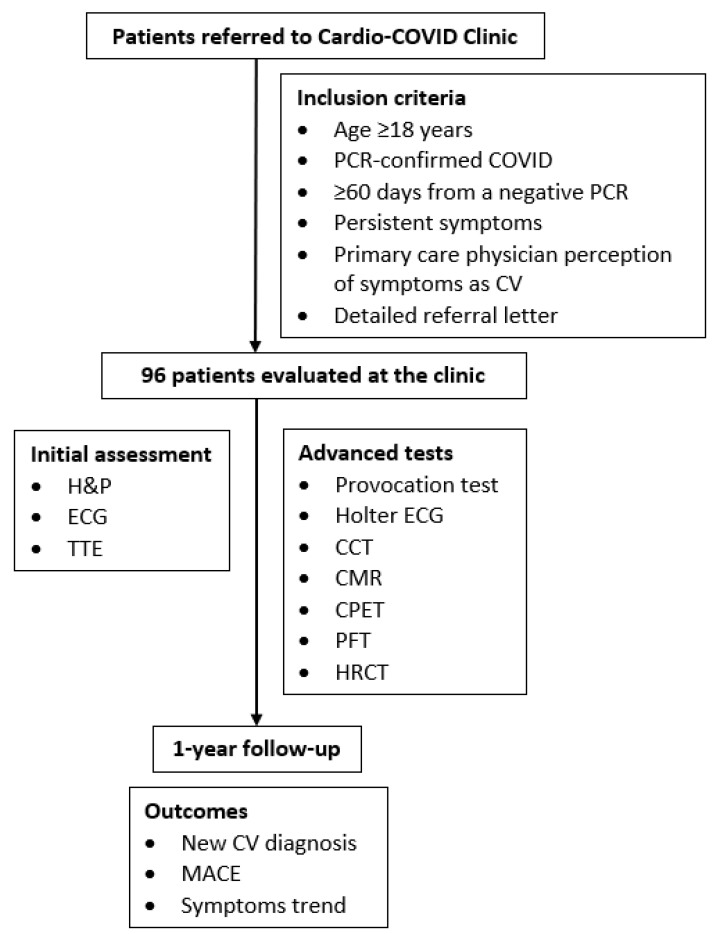
Study flow chart. CCT = cardiac computed tomography; CMR = cardiac magnetic resonance; COVID = coronavirus disease; CPET = cardiopulmonary exercise test; CV = cardiovascular; ECG = electrocardiogram; H&P = history and physical; HRCT = high-resolution CT; MACE = major adverse cardiovascular events; PCR = polymerase chain reaction; PFT = pulmonary function test; TTE = transthoracic echocardiogram.

**Figure 2 jcm-11-06123-f002:**
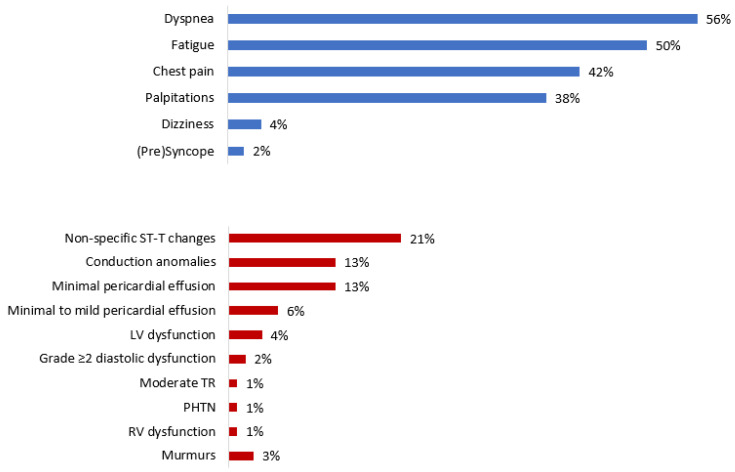
Presentation and objective findings demonstrated in Long-COVID Patients with suspected cardiovascular symptomatology. Among long-COVID outpatients assessed at a dedicated ‘Cardio’-COVID clinic due to suspected cardiovascular symptomatology, the leading symptoms at the initial visit were mainly non-specific, such as dyspnea and fatigue. Electrocardiographic aberrations were the most common objective findings and mainly included non-specific ST-T changes. COVID = coronavirus disease; CV = cardiovascular; LV = left ventricular; PHTN = pulmonary hypertension; RV = right ventricular; TR = tricuspid regurgitation.

**Figure 3 jcm-11-06123-f003:**
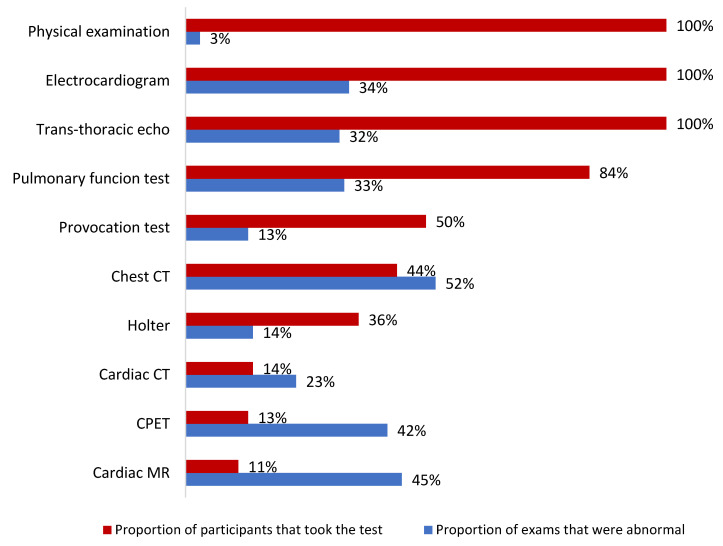
Diagnostic tests performed on the study cohort. While basic work-up proved negative in most patients examined for presumed cardiovascular sequelae, more advanced tests revealed pathologic findings at a higher frequency. Notably, most chest CT studies displayed anomalies. CPET = cardiopulmonary exercise test; CT = computed tomography; MR = magnetic resonance.

**Figure 4 jcm-11-06123-f004:**
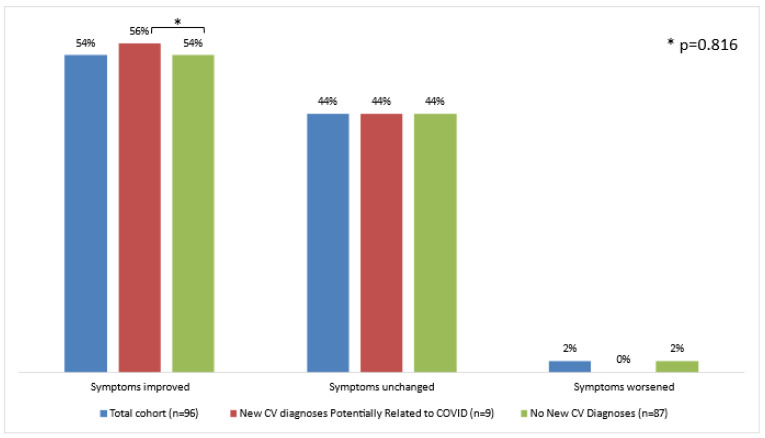
One-year trends of long-COVID symptomatology believed to be cardiovascular. At 1-year follow-up, most patients felt their symptoms to improve compared to the initial visit. No major difference was noted between those who were diagnosed with new, potentially COVID-related cardiovascular conditions and those who were not. COVID = coronavirus disease; CV = cardiovascular.

**Table 1 jcm-11-06123-t001:** Baseline patient characteristics.

	Total Cohort (*n* = 96)
Demographic Details	
Age, years	54 (44–64)
Male	44 (46)
Cardiovascular Background	
Prior CV Disease	15 (16)
CV Risk Factors	
Any	69 (72)
Pre-diabetes/Diabetes Mellitus	34 (35)
Hypertension	22 (23)
Dyslipidemia	56 (58)
Obesity	30 (31)
Present Smoking	10 (10)
Acute Covid Data	
Symptoms During Acute COVID	95 (99)
Acute COVID Severity, per NIH Criteria	
Mild	40 (42)
Moderate	30 (31)
Severe	26 (27)
Hospitalization	
Frequency	28 (29)
Length, days	8 (5–18)
Invasive Ventilation	1 (3.4)
Abnormal Tests Results	
ECG	10 (45)
TTE	2 (25)
Serum Biomarkers (hs-cTn ± NT-proBNP)	3 (13)
Time from COVID Diagnosis to Examination, days	142 (111–197)
COVID Vaccination Doses	
0	8 (10)
1	27 (35)
2	42 (55)

Data are presented as number (percentage) and median (interquartile range), where appropriate. COVID = coronavirus disease; CV = cardiovascular; ECG = electrocardiogram; hs-cTn = high-sensitivity cardiac troponin; NA = not applicable; NIH = National Institutes of Health; NT-proBNP = N-terminal-pro-brain natriuretic peptide; TTE = transthoracic echocardiogram.

**Table 2 jcm-11-06123-t002:** Long-COVID symptoms and objective findings.

	Total Cohort (*n* = 96)
Symptoms	
Any	96 (100)
Fatigue	48 (50)
Dyspnea	54 (56)
Chest Pain	40 (42)
Palpitations	36 (38)
Dizziness/Vertigo	4 (4)
Pre-syncope/Syncope	2 (2)
NYHA Class Functional Status	
I	43 (45)
II	41 (43)
III	12 (12)
IV	0 (0)
ECG Findings	
Any	32 (34)
Rhythm Disorders	3 (3)
Rate Disorders	0 (0)
Conduction Disorders	12 (13)
ST-T changes	20 (21)
TTE Findings	
Any	31 (32)
LV Systolic Dysfunction	4 (4)
RV Systolic Dysfunction	1 (1)
Grade2 and up Diastolic Dysfunction	2 (2)
LA Dilatation	6 (6)
Moderate and up Valvular Dysfunction	1 (1)
Systolic Pulmonary Hypertension	1 (1)
Pericardial Effusion	18 (19)
Cardiac Provocation Test Findings	
Any	6 (13)
Ischemic Findings	2 (4)
Atrial Fibrillation	1 (2)
Chronotropic Incompetence	3 (6)
Age-Adjusted CV Fitness	
Good	15 (39)
Average	14 (36)
Low	10 (26)
Holter ECG findings	
Any	5 (14)
Inappropriate Sinus Tachycardia	1 (3)
Atrial Tachycardia	1 (3)
Non-Sustained Ventricular Tachycardia	1 (3)
Couplet Ventricular Premature Beats	1 (3)
1st degree AV block	1 (3)
CCT findings	
Any	3 (23)
Non-Obstructive Coronary Artery Disease	2 (15)
Aberrant Coronary Artery	1 (8)
CMR findings	
Any	5 (45)
Myocarditis	3 (27)
Myopericarditis	1 (9)
Hypertrophic obstructive cardiomyopathy	1 (9)
CPET findings	
CV Restraint	5 (42)
PFT findings	
Any	27 (33)
Obstruction	13 (16)
Restriction	5 (6)
Reduced Diffusing Capacity	9 (11)
HRCT findings	
Any	22 (51)
Interstitial Changes	9 (21)
Fibrotic Changes	4 (10)
Ground Glass Opacities	7 (17)
Lung Nodules	2 (5)

Data are presented as number (percentage). CMR = cardiac magnetic resonance; COVID = coronavirus disease; CPET = cardiopulmonary exercise test; CCT = cardiac computed tomography; CV = cardiovascular; ECG = electrocardiogram; HRCT = high-resolution CT; LA = left atrium; LV = left ventricle; NA = not applicable; NYHA = New York Heart Association; PFT = pulmonary function test; TTE = transthoracic echocardiogram.

**Table 3 jcm-11-06123-t003:** Baseline patient characteristics according to the presence of a new, presumably COVID-related cardiovascular diagnosis.

	New CV Diagnoses (*n* = 9)	No New CV Diagnoses (*n* = 87)	*p*-Value
Demographic Details			
Age, years	52 (41–54)	56 (45–66)	0.139
Male	2 (22)	42 (48)	0.173
Cardiovascular Background			
Prior CV Disease	0 (0)	15 (17)	0.346
CV Risk Factors			
Any	5 (56)	64 (74)	0.263
Pre-diabetes/Diabetes Mellitus	1 (11)	33 (38)	0.152
Hypertension	2 (22)	20 (23)	0.949
Dyslipidemia	2 (22)	54 (62)	0.069
Obesity	4 (44)	26 (30)	0.454
Present Smoking	1 (11)	9 (10)	1.000
Acute COVID Data			
Symptoms During Acute COVID	9 (100)	86 (99)	1.000
Acute COVID Severity, per NIH Criteria			0.450
Mild	2 (22)	38 (44)	
Moderate	4 (44)	26 (30)	
Severe	3 (33)	23 (26)	
Hospitalization			
Frequency	2 (22)	26 (30)	1.000
Length, days	18 (17–20)	8 (5–18)	0.784
Invasive Ventilation	0 (0.0)	1 (4)	NA
Abnormal Tests Results			
ECG	1 (33)	9 (47)	1.000
TTE	1 (50)	1 (17)	0.464
Serum Biomarkers (hs-cTn and/or NT-proBNP)	1 (25)	2 (11)	0.453
Time from COVID Diagnosis to Examination, days	152 (75–214)	141 (112–195)	0.912
COVID Vaccination Doses			0.399
0	2 (25)	6 (9)	
1	2 (25)	25 (36)	
2	4 (50)	38 (55)	

Data are presented as number (percentage) and median (interquartile range), where appropriate. COVID = coronavirus disease; CV = cardiovascular; ECG = electrocardiogram; hs-cTn = high-sensitivity cardiac troponin; NA = not applicable; NIH = National Institutes of Health; NT-proBNP = N-terminal-pro-brain natriuretic peptide; TTE = transthoracic echocardiogram.

**Table 4 jcm-11-06123-t004:** Long-COVID symptoms and objective findings according to the presence of a new, presumably COVID-related cardiovascular diagnosis.

	New CV Diagnoses (*n* = 9)	No New CV Diagnoses (*n* = 87)	*p*-Value
Symptoms			
Any	8 (89)	86 (99)	1.000
Fatigue	7 (78)	41 (47)	0.159
Dyspnea	7 (78)	47 (54)	0.291
Chest Pain	4 (44)	36 (41)	1.000
Palpitations	6 (67)	30 (35)	0.076
Dizziness/Vertigo	1 (11)	3 (3)	0.330
Pre-syncope/Syncope	0 (0)	2 (2)	1.000
NYHA Class Functional Status			0.444
I	4 (44)	39 (45)	
II	5 (56)	36 (41)	
III	0 (0)	12 (14)	
IV	0 (0)	0 (0)	
Symptom Non-Improvement	4 (44)	40 (46)	0.816
ECG Findings			
Any	4 (44)	28 (32)	0.475
Rhythm Disorders	0 (0)	3 (3)	1.000
Rate Disorders	0 (0)	0 (0)	1.000
Conduction Disorders	2 (22)	10 (11)	0.319
ST-T changes	3 (33)	17 (20)	0.387
TTE Findings			
Any	4 (44)	27 (31)	0.471
LV Systolic Dysfunction	1 (11)	3 (3)	0.246
RV Systolic Dysfunction	0 (0)	1 (1)	1.000
Grade2 and up Diastolic Dysfunction	0 (0)	2 (2)	0.863
LA Dilatation	2 (22)	4 (5)	0.017
≥Moderate Valvular Dysfunction	1 (11)	0 (0)	1.000
Systolic Pulmonary Hypertension	1 (11)	0 (0)	0.700
Pericardial Effusion	2 (22)	16 (18)	0.124
Cardiac Provocation Test Findings			
Any	3 (50)	3 (7)	0.200
Ischemic Findings	0 (0)	2 (5)	1.000
Atrial Fibrillation	0 (0)	1 (2)	1.000
Chronotropic Incompetence	3 (50)	0 (0)	1.000
Age-Adjusted CV Fitness			0.453
Good	1 (17)	14 (42)	
Average	2 (33)	12 (36)	
Low	3 (50)	7 (21)	
Holter ECG findings			0.139
Any	2 (40)	3 (10)	
Inappropriate Sinus Tachycardia	1 (20)	0 (0)	
Atrial Tachycardia	1 (20)	0 (0)	
Non-Sustained Ventricular Tachycardia	0 (0)	1 (3)	
Couplet Ventricular Premature Beats	0 (0)	1 (3)	
1st degree AV block	0 (0)	1 (3)	
CCT findings			1.000
Any	0 (0)	3 (27)	
Non-Obstructive Coronary Artery Disease	0 (0)	2 (18)	
Aberrant Coronary Artery	0 (0)	1 (9)	
CMR findings			0.242
Any	4 (67)	1 (20)	
Myocarditis	3 (50)	0 (0)	
Myopericarditis	1 (17)	0 (0)	
Hypertrophic obstructive cardiomyopathy	0 (0)	1 (20)	
CPET findings			
CV Restraint	2 (100)	3 (30)	0.152
PFT findings			
Any	6 (67)	21 (29)	0.054
Obstruction	5 (56)	8 (11)	0.084
Restriction	0 (0)	5 (69)	0.076
Reduced Diffusing Capacity	1 (11)	8 (11)	0.243
HRCT findings			
Any	3 (43)	19 (54)	0.691
Interstitial Changes	1 (14)	8 (23)	0.362
Fibrotic Changes	1 (14)	3 (9)	0.846
Ground Glass Opacities	0 (0)	7 (20)	0.124
Lung Nodules	1 (14)	1 (3)	0.632

Data are presented as number (percentage). CMR = cardiac magnetic resonance; COVID = coronavirus disease; CPET = cardiopulmonary exercise test; CCT = cardiac computed tomography; CV = cardiovascular; ECG = electrocardiogram; HRCT = high-resolution CT; LA = left atrium; LV = left ventricle; NA = not applicable; NYHA = New York Heart Association; PFT = pulmonary function test; TTE = transthoracic echocardiogram.

**Table 5 jcm-11-06123-t005:** Univariate binary logistic regression model for the outcomes of new, potentially COVID-related cardiovascular diagnoses and symptom non-improvement.

	New, Potentially COVID-Related CV Diagnoses		Symptom Non-Improvement	
	OR (95% CI)	*p* Value	OR (95% CI)	*p* Value
Baseline Clinical Variables				
Age (continuous)	0.96 (0.92–1.01)	0.144	1.01 (0.97–1.04)	0.755
Sex Male	0.31 (0.06–1.56)	0.154	1.04 (0.39–2.76)	0.939
Prior CV Disease	0.90 (0.86–1.23)	0.655	0.38 (0.09–1.57)	0.179
CV Risk Factors	0.45 (0.11–1.82)	0.262	1.31 (0.43–4.03)	0.632
Acute COVID Parameters				
Severity				
Moderate vs. Mild	2.92 (0.50–17.15)	0.235	1.40 (0.54–3.61)	0.491
Severe vs. Mild	2.48 (0.39–15.96)	0.340	0.76 (0.28–2.09)	0.600
≥Moderate vs. Mild	2.71 (0.53–13.82)	0.229	0.68 (0.25–1.86)	0.456
Severe vs. Non-Severe	1.39 (0.32–6.02)	0.659	0.82 (0.29–2.34)	0.713
Hospitalization	0.67 (0.13–3.45)	0.632	0.49 (0.17–1.39)	0.177
Hospitalization Length	1.11 (0.93–1.34)	0.251	0.93 (0.82–1.05)	0.218
Abnormal In-Hospital ECG	0.56 (0.04–7.21)	0.653	5.33 (0.62–45.99)	0.128
Abnormal In-Hospital TTE	5.00 (0.15–166.59)	0.368	2.00 (0.05–78.25)	0.711
Abnormal In-Hospital Cardiac Biomarkers	2.83 (0.19–41.99)	0.449	0.57 (0.04–7.74)	0.674
COVID Vaccination Status				
COVID Vaccination	0.29 (0.05–1.74)	0.174	1.57 (0.24–10.24)	0.640
Long COVID Presentation				
Fatigue	3.93 (0.77–19.98)	0.099	2.00 (0.74–5.39)	0.170
Dyspnea	2.98 (0.59–15.16)	0.189	1.75 (0.63–4.90)	0.286
Chest Pain	1.13 (0.28–4.52)	0.859	1.96 (0.73–5.28)	0.182
Palpitations	3.8 (0.89–16.23)	0.072	1.68 (0.62–4.56)	0.311
Dizziness/Vertigo	3.50 (0.33–37.69)	0.302	0.79 (0.33–4.28)	0.673
Pre-syncope/Syncope	2.21 (0.89–10.23)	0.872	3.21 (0.05–5.48)	0.915
NYHA Class Functional Status				
Continuous	0.53 (0.16–1.74)	0.294	2.38 (1.26–4.49)	0.008
Class II-III vs. I	0.59 (0.14–2.50)	0.472	2.70 (1.17–6.25)	0.020
Class III vs. I-II	1.00 (0.99–1.01)	0.999	4.20 (1.06–16.64)	0.041
Work-Up Findings				
Heart Murmur	0.67 (0.40–2.89)	0.203	1.17 (0.07–19.59)	0.912
Abnormal ECG	1.69 (0.41–6.94)	0.473	0.50 (0.14–1.76)	0.280
Abnormal TTE	1.72 (0.43–6.91)	0.446	2.84 (0.94–8.56)	0.064
LVEF (continuous)	0.96 (0.84–1.11)	0.582	0.98 (0.89–1.09)	0.737
LV Systolic Dysfunction	1.02 (0.24–19.53)	0.534	0.76 (0.32–4.54)	0.863
LA Dilatation	2.71 (0.48–15.34)	0.259	1.67 (0.26–10.74)	0.591
LAVi (continuous)	0.97 (0.89–1.12)	0.940	1.00 (0.93–1.08)	0.976
More than Mild Valvular Dysfunction	2.00 (0.21–19.23)	0.549	4.92 (0.52–46.78)	0.165
PASP (continuous)	0.91 (0.75–1.11)	0.368	1.18 (0.97–1.45)	0.104
Pericardial Effusion	2.33 (0.52–10.39)	0.266	6.64 (1.3–33.88)	0.023
Abnormal Provocation Test *			0.17 (0.02–1.71)	0.133
Average/Low (vs Good) Age-Adjusted CV Fitness	3.68 (0.39–35.14)	0.257	0.90 (0.21–3.82)	0.886
CV Restraint per CPET	2.89 (0.86–2.56)	0.811	0.75 (0.06–8.83)	0.819
Abnormal Holter ECG	NA	NA	0.97 (0.64–3.42)	0.723
Abnormal CCT	NA	NA	1 (0.06–15.99)	1.000
Abnormal CMR	NA	NA	0.44 (0.04–5.58)	0.530
Abnormal PFT	4.86 (1.11–21.26)	0.036	2.26 (0.75–6.80)	0.148
Abnormal HRCT	0.63 (0.12–3.25)	0.582	0.29 (0.07–1.22)	0.091
Outcomes				
New CV Diagnosis	NA	NA	1.19 (0.27–5.24)	0.816
Symptom Non-Improvement	1.19 (0.27–5.24)	0.816	NA	NA

* Abnormal provocation test was explored in regard to myocardial diagnoses only, as its findings included chronotropic incompetence. CI = confidence interval; CMR = cardiac magnetic resonance; COVID = coronavirus disease; CPET = cardiopulmonary exercise test; CCT = cardiac computed tomography; CV = cardiovascular; ECG = electrocardiogram; HRCT = high-resolution CT; LA = left atrium; LAVi = LA volume index; LV = left ventricle; LVEF = LV ejection fraction; NYHA = New York Heart Association; OR = odds ratio; PASP = pulmonary arterial systolic pressure; PFT = pulmonary function test; RV = right ventricle; TTE = transthoracic echocardiogram.

**Table 6 jcm-11-06123-t006:** Multivariate binary logistic regression model for the outcomes of new, potentially COVID-related cardiovascular diagnoses and symptom non-improvement.

	New, Potentially COVID-Related CV Diagnoses		Symptom Non-Improvement	
	OR (95% CI)	*p* Value	OR (95% CI)	*p* Value
Fatigue	2.89 (0.50–16.72)	0.235		
Palpitations	2.60 (0.53–12.73)	0.240		
NYHA Class Functional Status				
Continuous			1.76 (0.60–5.14)	0.303
Class II-III vs. I			1.55 (0.40–6.06)	0.530
Class III vs. I-II			3.97 (0.34–46.24)	0.271
Pericardial Effusion				
NYHA Class Continuous			2.26 (0.40–12.70)	0.353
NYHA Class II-III vs. I			2.40 (0.43–13.31)	0.316
NYHA Class III vs. I-II			1.92 (0.33–11.16)	0.470
Abnormal PFT	5.16 (1.12–23.68)	0.035		
Abnormal HRCT				
NYHA Class Continuous			1.17 (0.26–4.79)	0.882
NYHA Class II-III vs. I			0.98 (0.24–4.02)	0.974
NYHA Class III vs. I-II			1.17 (0.27–4.78)	0.872

CI = confidence interval; COVID = coronavirus disease; CV = cardiovascular; HRCT = high-resolution computed tomography; LVEF = LV ejection fraction; NYHA = New York Heart Association; OR = odds ratio; PFT = pulmonary function test.

**Table 7 jcm-11-06123-t007:** Baseline patient characteristics according to symptomatic course.

	Symptom Improvement (*n* = 52)	Symptom Non-Improvement (*n* = 44)	*p*-Value
Demographic Details			
Age, years	53 (42–66)	57 (45–62)	0.517
Male	22 (42)	22 (50)	0.451
Cardiovascular Background			
Prior CV Disease	10 (19)	5 (11)	0.290
CV Risk Factors			
Any	34 (65)	35 (80)	0.124
Pre-diabetes/Diabetes Mellitus	19 (37)	15 (34)	0.803
Hypertension	9 (17)	13 (30)	0.155
Dyslipidemia	30 (58)	26 (59)	0.890
Obesity	17 (33)	13 (30)	0.740
Present Smoking	3 (9)	7 (16)	0.178
Acute Covid Data			
Symptoms During Acute COVID	52 (100)	43 (98)	0.458
Acute COVID Severity, per NIH Criteria			0.533
Mild	22 (23)	18 (41)	
Moderate	14 (27)	16 (36)	
Severe	16 (31)	10 (23)	
Hospitalization			
Frequency	18 (35)	10 (23)	0.202
Length, days	11 (5–19)	7 (5–16)	0.586
Invasive Ventilation	1 (6)	0 (0)	NA
Abnormal Tests Results			
ECG	4 (33)	6 (60)	0.391
TTE	1 (25)	1 (25)	1.000
Serum Biomarkers (hs-cTn and/or NT-proBNP)	2 (15)	1 (10)	0.704
Time from COVID Diagnosis to Examination, days	140 (105–179)	143 (113–214)	0.405
COVID Vaccination Doses			0.314
0	6 (15)	2 (5)	
1	12 (30)	15 (41)	
2	22 (55)	20 (54)	

Data are presented as number (percentage) and median (interquartile range), where appropriate. COVID = coronavirus disease; CV = cardiovascular; ECG = electrocardiogram; hs-cTn = high-sensitivity cardiac troponin; NA = not applicable; NIH = National Institutes of Health; NT-proBNP = N-terminal-pro-brain natriuretic peptide; TTE = transthoracic echocardiogram.

**Table 8 jcm-11-06123-t008:** Long-COVID symptoms and objective findings according to symptomatic course.

	Symptom Improvement (*n* = 52)	Symptom Non-Improvement (*n* = 44)	*p*-Value
Symptoms			
Any	52 (100)	44 (100)	1.000
Fatigue	22 (42)	26 (59)	0.101
Dyspnea	27 (52)	27 (61)	0.353
Chest Pain	20 (39)	20 (46)	0.489
Palpitations	18 (35)	18 (41)	0.526
Dizziness/Vertigo	0 (0)	4 (9)	0.041
Pre-syncope/Syncope	1 (1)	1 (2)	0.207
NYHA Class Functional Status			0.022
I	29 (56)	14 (32)	
II	20 (39)	21 (48)	
III	3 (6)	9 (21)	
IV	0 (0)	0 (0)	
ECG Findings			
Any	17 (33)	15 (34)	0.658
Rhythm Disorders	2 (4)	1 (2)	0.498
Conduction Disorders	6 (12)	6 (14)	0.871
ST-T changes	13 (25)	8 (18)	0.387
TTE Findings			
Any	14 (27)	17 (39)	0.274
LV Systolic Dysfunction	3 (6)	1 (2)	0.245
RV Systolic Dysfunction	0 (0)	1 (2)	0.468
Grade2 and up Diastolic Dysfunction	1 (2)	1 (2)	1.000
LA Dilatation	3 (6)	3 (6)	0.509
≥Moderate Valvular Dysfunction	1 (2)	0 (0)	1.000
Systolic Pulmonary Hypertension	0 (0)	1 (2)	0.471
Pericardial Effusion	6 (12)	12 (27)	0.060
Cardiac Provocation Test Findings			
Any	4 (17)	2 (8)	0.666
Ischemic Findings	1 (4)	1 (4)	1.000
Atrial Fibrillation	0 (0)	1 (4)	0.700
Chronotropic Incompetence	3 (12)	0 (0)	0.121
Age-Adjusted CV Fitness			0.831
Good	9 (43)	6 (33)	
Average	7 (33)	7 (39)	
Low	5 (24)	5 (28)	
Holter ECG findings			0.141
Any	1 (5)	4 (27)	
Inappropriate Sinus Tachycardia	1 (5)	0 (0)	
Atrial Tachycardia	0 (0)	1 (7)	
Non-Sustained Ventricular Tachycardia	0 (0)	1 (7)	
Couplet Ventricular Premature Beats	0 (0)	1 (7)	
1st degree AV block	0 (0)	1 (7)	
CCT findings			1.000
Any	1 (25)	2 (22)	
Non-Obstructive Coronary Artery Disease	0 (0)	2 (22)	
Aberrant Coronary Artery	1 (25)	0 (0)	
CMR findings			1.000
Any	3 (50)	2 (40)	
Myocarditis	2 (33)	1 (20)	
Myopericarditis	0 (0)	1 (20)	
Hypertrophic obstructive cardiomyopathy	1 (17)	0 (0)	
CPET findings			
CV Restraint	2 (40)	3 (43)	1.000
PFT findings			
Any	12 (27)	15 (42)	0.155
Obstruction	4 (9)	9 (25)	0.168
Restriction	3 (7)	2 (6)	0.628
Reduced Diffusing Capacity	5 (11)	4 (12)	0.448
HRCT findings			
Any	13 (59)	9 (45)	0.361
Interstitial Changes	6 (27)	3 (15)	0.346
Fibrotic Changes	2 (9)	2 (10)	1.000
Ground Glass Opacities	4 (18)	3 (15)	0.876
Lung Nodules	1 (5)	1 (5)	1.000
New, Potentially COVID-Related CV Diagnosis	5 (10)	4 (9)	1.000

Data are presented as number (percentage). CMR = cardiac magnetic resonance; COVID = coronavirus disease; CPET = cardiopulmonary exercise test; CCT = cardiac computed tomography; CV = cardiovascular; ECG = electrocardiogram; HRCT = high-resolution CT; LA = left atrium; LV = left ventricle; NA = not applicable; NYHA = New York Heart Association; PFT = pulmonary function test; TTE = transthoracic echocardiogram.

## Data Availability

The data underlying this article will be shared on reasonable request to the corresponding author.

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
