# Peer review of "Assessment of Adult Patients with Long COVID Manifestations Suspected as Cardiovascular: A Single-Center Experience"

_jcm, 2022, doi:10.3390/jcm11206123_

Round 1
Reviewer 1 Report
Major comment:
The authors propose a study where they investigate the specific burden of cardiovascular morbidity in long COVID patients. While the study methodology is appropriate and the objectives/research question are of interest to the scientific community, as also acknowledged by the authors, the study is strongly underpowered (total of 96 patients with only 9 diagnoses of “never-before known, potentially COVID-related, cardiovascular” diseases). This limitation questions the validity of the conclusions drawn from the study; i.e., 1) the differences (or the absence of differences) between groups reported by the authors may be due to the lack of statistical power, and 2) the limited group size may not be representative of a general population of long COVID patients. Given this major limitation, the statement the authors provide in the Method section (lines 77-78, “While the clinic is still running, this report relates to patients seen at the clinic between June 2020 and June 2021”) is somewhat surprising, as an additional year worth of data may considerably improve the value of the study. Why were patients who attended the clinic from June 2021 to June 2022 not included in the study?
Other comments:
1. Abstract, lines 25-27 – The first sentence of the abstract is unclear. Please consider revising/rephrasing.
2. Abstract, lines 37 – Do the author refer to 142 days after the diagnosis of COVID or of CV symptoms? Please clarify.
3. Introduction, line 53 – I am not familiar with the term “myocardial invasion”.
4. Introduction, line 56 – The authors state that long COVID “may affect up to nearly 90% of survivors”. This percentage seems quite high and surprising. Are you referring to a general population?
5. Methods, line 69 – Please add “(city, country)” after Rabin Medical Center (RMC), Beilinson Hospital’s Cardio-COVID Clinic.
6. Results, lines 143-144 – The authors state that “Late symptoms not known to affect patients prior to the infectious diseases were present in 94 (98%) cases”. This sentence suggest that 2 people were included in the study despite not matching the inclusion criteria (Methods, lines 70-71: “adults […] patients recovering from COVID-19 who manifested ongoing signs and/or symptoms suspected by their treating physician to represent CV sequelae”). Where these patients already affected by the CV symptoms before contracting COVID-19 or not affected by CV symptoms at all? Please clarify the reasons for inclusion.
Reviewer 2 Report
Shechter et al try to describe the cardiovascular symptoms, diagnoses and 1-year follow up complications of a cohort of long COVID patients. They apparently suggest that only a small minority of these patients have cardiovascular diagnoses, and that regardless of the specific diagnoses they do not have major cardiovascular complications during 1-year follow up period.
Following are my major comments:
1. The most troublesome issue is the relatively small number of patients who underwent advanced imaging, and specifically MRI. Thus, the authors suggest there were only 5 patients with myocarditis or perimyocarditis; however, this is the result of the small numbers who underwent MRI. Indeed, the authors say that even among those cases, there were some who had normal echo, emphasizing the importance of MRI in such diagnoses, which at times will not have any clues on regular TTE. As for the arrhythmia group diagnoses, only 1/3 underwent 24-hour Holter recordings and thus, again, there is by definition underestimation of the true prevalence of such diagnoses among these patients. Accordingly, the main aim of the study which was to assess the burden of cardiovascular diagnoses in these group is apparently an underestimation of the true prevalence. Thus, I would suggest speaking more on the cardiovascular symptoms and the follow up outcomes of these patients regardless of the true specific diagnosis.
2. As a continuation of the above comment the paper text should be revised. For example, I feel it is misleading to say that 45% of MRI revealed pathologies, if only 11/96 had MRI done. Moreover, I am not sure if any sense looking for multivariate analysis for having CV diagnosis, given the significant underestimation which is highly probable. The authors could however suggest that there were few if any severe myocarditis cases by clinical judgement.
3. Regarding the favorable prognosis of these patients- this was based on absence of MACE events including ACS, stroke or death. However, the fact that almost half of patients did not improve their symptoms should be emphasized as well. Here again, it is very probable that the absence of predictors for no improvement is a result of the small number of patients who underwent MRI. If extrapolating from the prior non-COVID related myocarditis studies, I would expect MRI to have at least some predicting value. As for the arrhythmia group- not clear if these patients underwent any further studies, like repeat Holter, during their follow up period. If for example we speak on AF diagnosis, I am not sure that absence of stroke within 1 year implies a good prognosis. More importantly, would like to know how many of these patients continued with repeat episodes of AF.
Minor comments
1. Recommend adding the physical examination finding in Table 1 and if possible, the CK and/or Troponin levels during acute COVID, to see if they could predict future myocarditis, which might have gone asymptomatic for a while.
2. Would be nice to know what were the criteria according to which 26 patients were defined as ‘severe COVID”, if only 1 was ventilated. Were all the other hemodynamically unstable requiring ionotropic support? Other reasons?

Reviewer 3 Report
Dear Authors,
The work is original, intriguing... I recommend the right layout then I could suggest a title? Long COVID-19 syndromes mimicking cardiovascular disease
Abstract
conventionally before the background and after the goal
In the methods put a draft of inclusion criteria of the population of interest. The concept of COVID cardio is not a method but it is background material. In population, intervention and outcome methods
41-42 what test did you use to say relationship
54-57 42-44 Better as sort of << However, COVID-19 might have detrimental sequelae even after the post-acute phase, depicting a new pathological condition: the “post-COVID-19 syndrome (PCS)” or “long COVID” >> reference: https://www.mdpi.com/2076-3417/12/17/8593
As a goal? The study aimed to evaluate how many CVD-mimicking long COVID syndromes actually had an organic cardiovascular nature (?)
68 Report city and state
71 there are references to this structure, have other manuscripts been produced?
72 fate una sezione popolazione divisa dal design, seguite lo STROBE checklist..
Participants --- Give the eligibility criteria, and the sources and methods of selection of participants
Variables --- Clearly define all outcomes, exposures, predictors, potential confounders, and effect modifiers. Give diagnostic criteria, if applicable.
in this regard you have only mentioned impending events, which among other things denature the transversal study being of a prospective nature. if the study is transversal .. have these imitators carried out some functional scale? the fatigue? some score related to CVD markers
Bias --- Describe any efforts to address potential sources of bias
In the statistical analysis certain assessments are described, but when it comes to the odd ratio there is no calculation of the relative risk
In Table 1, however, you should define how you are going to stratify the groups ...
In addition, I would reformulate table 3, eliminating those empty spaces that make the results lose soundness
Round 2
Reviewer 1 Report
Thank you for answering my question and comments.
Reviewer 3 Report
Dear authors, the manuscript is clearer, but I am not convinced of the methodological approach:
245 “Symptoms of possible CV nature were evident well after formal recovery from COVID and mostly comprised of less-specific ones (such as dyspnea 246 and fatigue)”
Was the CV syndrome evident? but it included nonspecific symptoms of dyspnea and fatigue… What were the inclusion criteria? They seem a bit too large and unclear in the methods. why have not you evaluated a BORG cut-off (for example)? Because if the eligibility of the sample is not clear, then how do you say there is a low prevalence of organic CV disorders in Long COVID-19. Unfortunately, at this point your level of suspicion affects the prevalence of CV too much.
Other minor concerns
Table 2, does the p value refer to X2?
165 is there any correlation between severity and mimicking CV?
